# Microbial Valorization of Agricultural and Agro-Industrial Waste into Bacterial Cellulose: Innovations for Circular Bioeconomy Integration

**DOI:** 10.3390/microorganisms13122686

**Published:** 2025-11-25

**Authors:** Ayaz M. Belkozhayev, Arman Abaildayev, Bekzhan D. Kossalbayev, Kuanysh T. Tastambek, Danara K. Kadirshe, Gaukhar Toleutay

**Affiliations:** 1Department of Chemical and Biochemical Engineering, Geology and Oil-Gas Business Institute Named After K. Turyssov, Satbayev University, Almaty 050043, Kazakhstan; a.abaildayev@satbayev.university (A.A.); kossalbayev.bekzhan@gmail.com (B.D.K.); 2Sustainability of Ecology and Bioresources, Al-Farabi Kazakh National University, Al-Farabi Ave. 71, Almaty 050040, Kazakhstan; tastambeku@gmail.com; 3Ecology Research Institute, Khoja Akhmet Yassawi International Kazakh Turkish University, Turkistan 161200, Kazakhstan; 4International Faculty, Asfendiyarov Kazakh National Medical University, Almaty 050012, Kazakhstan; kadirwe.d@kaznmu.kz; 5Department of Chemistry, University of Tennessee, Knoxville, TN 37996, USA

**Keywords:** microbial valorization, bacterial cellulose, agricultural waste, sustainable bioprocesses, circular bioeconomy

## Abstract

Agricultural and agro-industrial waste, produced in vast quantities worldwide, presents both environmental and economic challenges. Microbial valorization offers a sustainable solution, with bacterial cellulose (BC) emerging as a high-value product due to its purity, strength, biocompatibility, and biodegradability. This review highlights recent advances in producing BC from agricultural and agro-industrial residues via optimized fermentation processes, including static and agitated cultivation, co-cultivation, stepwise nutrient feeding, and genetic engineering. Diverse wastes such as fruit peels, sugarcane bagasse, cereal straws, and corn stover serve as cost-effective carbon sources, reducing production costs and aligning with circular bioeconomy principles. Advances in strain engineering, synthetic biology, and omics-guided optimization have significantly improved BC yield and functionalization, enabling applications in food packaging, biomedicine, cosmetics, and advanced biocomposites. Process innovations, including tailored pretreatments, adaptive evolution, and specialized bioreactor designs, further enhance scalability and product quality. The integration of BC production into circular bioeconomy models not only diverts biomass from landfills but also replaces petroleum-based materials, contributing to environmental protection and resource efficiency. This review underscores BC’s potential as a sustainable biomaterial and identifies research directions for overcoming current bottlenecks in industrial-scale implementation.

## 1. Introduction

Agricultural production worldwide generates vast quantities of waste annually, much of which remains unprocessed. A large portion of this agricultural and agro-industrial waste is either landfilled or incinerated, practices that not only pose environmental hazards but also significantly contribute to greenhouse gas emissions [1,2]. In accordance with recent estimates from WasteManaged UK, the agricultural sector generates approximately 1.3 to 2.1 billion tons of waste annually worldwide, including crop residues, livestock manure, and agro-industrial by-products [3]. In the European Union, agricultural biomass accounts for around 900 million tons per year, a significant portion of which consists of crop residues and other underutilized by-products from farming activities [4]. In Asia, particularly in countries like China and India, agricultural and agro-industrial waste generation is estimated to exceed 1.5 billion tons per year, largely due to extensive cereal and rice production [5,6,7]. In this context, the effective management of agricultural and agro-industrial waste through recycling and conversion into value-added products has become one of the pressing priorities of modern science and industry [8,9]. Utilizing agro-industrial waste not only contributes to environmental protection but also holds the potential for generating additional economic benefits. This approach contributes to addressing the problem of agricultural waste disposal, reducing soil degradation, and decreasing greenhouse gas emissions [10,11].

Bacterial cellulose (BC) is a highly pure, renewable nanopolymer synthesized by certain prokaryotes [12]. It exhibits excellent mechanical strength, water retention, and biocompatibility. Unlike plant cellulose, BC is free from lignin and hemicellulose, allowing easier modification for various applications. These attributes make BC a promising material for research and industrial use [13,14]. The microorganisms primarily known for the biosynthesis of BC are Gram-negative acetic acid bacteria [15]. In particular, members of the genus *Komagataeibacter* (formerly classified as *Acetobacter* or *Gluconacetobacter*) are well recognized for their high cellulose-producing capacity at the air–liquid interface of the culture medium [16]. In recent years, increasing attention has been given to the use of low-cost agro-industrial waste as an alternative raw material to replace expensive refined sugars in BC production [17]. For example, residues from plant-based and food processing by-products such as fruit pulps, vegetable waste, and cereal straws can serve as viable carbon sources for cellulose-producing bacteria [18,19]. Utilizing agricultural and agro-industrial waste in this manner addresses two major challenges simultaneously: enhancing economic efficiency by reducing production costs and contributing to environmental sustainability by recycling otherwise unusable biomass [20,21,22]. To achieve efficient industrial-scale production of BC, optimization of the production process remains a critical challenge [23]. Although the currently used static cultivation method yields high-quality cellulose, it suffers from low productivity and prolonged cultivation times [24]. In contrast, agitated cultivation techniques offer faster production rates but may result in inconsistent product quality. Additionally, the high cost of nutrient media and the genetic instability of bacterial strains pose significant barriers to large-scale commercialization [25,26]. To address these challenges, researchers are exploring innovative strategies such as the use of low-cost agro-industrial waste as substrates, pulse-feeding systems, co-fermentation techniques, and genetic engineering approaches [27,28,29]. BC is a biodegradable, eco-friendly material used across food packaging, medicine, cosmetics, textiles, and electronics [30]. Its effectiveness in preserving food and natural decomposition make it a sustainable alternative to plastic, positioning it as a promising green innovation for the food industry [31,32]. The production of BC from agricultural and agro-industrial waste represents a sustainable solution that fully aligns with the principles of the circular bioeconomy [33,34]. This model transforms waste into a valuable resource, reduces environmental burdens, and paves the way for the development of emerging sectors such as biopolymer production. In addition to BC-focused valorization, agro-industrial residues can also be converted into other key bioproducts. Marchetti et al. (2025) [35] showed that reground pasta, a carbohydrate-rich food by-product, can be efficiently transformed into volatile fatty acids (VFAs) through mixed-culture acidogenic fermentation, achieving ~54% COD conversion with effluents enriched in acetic, butyric, and propionic acids. These VFAs represent important precursors for downstream biopolymer synthesis. Complementary to this, Marchetti et al. (2024) [36] demonstrated that VFA-rich fermented pasta streams can serve as an effective carbon source for polyhydroxyalkanoate (PHA) production by phototrophic purple bacteria, reaching hydroxyvalerate contents of up to 60% in a semi-continuous photo-bioreactor. Together, these studies illustrate that food-industry by-products can support multiple microbial valorization pathways, broadening the portfolio of waste-derived bioproducts within circular bioeconomy frameworks. By integrating agriculture and biotechnology, BC production is becoming a key component of the green economy [37,38]. Overall, the production of BC from agricultural and agro-industrial waste represents a promising direction based on green technologies, offering both environmental sustainability and economic efficiency. In contrast to earlier reviews, the present work explicitly focuses on agro-industrial residues as feedstocks and links process innovations with circular bioeconomy perspectives, thus providing a systems-level view not previously covered.

This review article provides a comprehensive overview of the structure and biosynthesis of BC, the potential use of agricultural and agro-industrial waste as feedstock, strategies for optimizing the production process, and its diverse application areas. Additionally, the article explores the prospects for integrating this technology into the circular bioeconomy and outlines key directions for future research. In this review, the literature was identified through targeted searches in the Web of Science, Scopus, PubMed, ScienceDirect, and Google Scholar databases, with a focus on recent publications. The search used combinations of keywords such as “bacterial cellulose”, “bacterial nanocellulose”, “*Komagataeibacter*”, “agricultural waste”, “agro-industrial residues”, “waste valorization”, and “circular bioeconomy”. Studies were included if they addressed BC biosynthesis, the use of agricultural or agro-industrial residues as substrates, or the properties and applications of BC, and were available as full-text scientific publications. Although recent studies were prioritized, several earlier seminal papers were retained because they provide foundational insights that remain essential for understanding BC biosynthesis and its mechanistic background.

## 2. Biosynthesis and Cellulose-Producing Microorganisms

### 2.1. The Biosynthesis Process of BC

BC forms a characteristic extracellular nanofibrous network produced by *Komagataeibacter* species. Its biosynthesis is a microbially driven process that integrates central carbohydrate metabolism with the activity of specialized cellulose synthase complexes. The pathway begins when a carbohydrate substrate (e.g., glucose) enters the cell and is phosphorylated by glucokinase, forming glucose-6-phosphate (G6P), the initial committed step in BC synthesis [39,40]. G6P is then converted into glucose-1-phosphate (G1P) by phosphoglucomutase [41]. In the next stage, UDP-glucose pyrophosphorylase (UGPase, encoded by *galU*) catalyzes the reaction between G1P and uridine triphosphate (UTP), producing uridine diphosphate-glucose (UDP-glucose) [42]. This activated sugar nucleotide serves as the direct precursor for polymerization into β-1,4-glucan chains, and its pool size is a key determinant of BC yield. UGPase activity in cellulose-producing bacteria is ~100-fold higher than in non-cellulose-producing strains, underscoring its central role in the pathway [43,44]. The final stage involves the stepwise addition of UDP-glucose monomers by the cellulose synthase complex (*bcs* complex), followed by extracellular assembly into cellulose fibrils [45,46]. The core catalytic subunit BcsA is embedded in the inner membrane and contains a glycosyltransferase (GT-2) active site for glucan chain elongation. Its PilZ domain binds the bacterial second messenger cyclic-di-GMP, an allosteric activator that tightly couples BC synthesis to intracellular signaling levels [47,48]. BcsB, located in the periplasm, partners with BcsA to translocate the growing glucan chain across the periplasmic space [49]. BcsC forms an outer membrane pore for cellulose export, structurally resembling type II/IV secretion channel proteins [50,51]. BcsD, although non-essential for polymerization, enhances chain alignment and crystallization; deletion of *bcsD* can reduce BC output by up to 90% [44,52,53]. Together, these subunits orchestrate the continuous extrusion of cellulose microfibrils, which spontaneously assemble into the characteristic nanofibrous BC network at the cell surface (Figure 1).

After secretion, β-1,4-glucan chains self-assemble into protofibrils (~2–4 nm), which aggregate into ribbon-like microfibrils (~80 nm) and form a three-dimensional network stabilized by hydrogen bonds and van der Waals forces [54,55,56]. This produces the characteristic white BC pellicle at the air–liquid interface. Members of the genus *Komagataeibacter* grow aerobically, forming cellulose films that enhance surface adhesion; biofilm stability; and protection against UV, pH shifts, and toxins [57,58,59]. Under static conditions, BC accumulates at oxygen-rich surfaces; agitation increases oxygen distribution but disrupts film formation, yielding dispersed granules and often lowering productivity [60,61,62].

### 2.2. Microorganisms That Synthesize BC

Prokaryotic cellulose producers span multiple bacterial taxa. The primary and most industriously exploited group are Gram-negative Proteobacteria, especially acetic acid bacteria of the genus *Komagataeibacter* [63]. Other contributors include *Rhizobium*, *Agrobacterium*, *Enterobacter*, and *Pseudomonas*, as well as rare cases of Gram-positive Sarcina ventriculi [63,64]. Among these, *Komagataeibacter xylinus* remains the model organism due to its ability to convert sugars (e.g., glucose, fructose, sucrose, mannitol) into large volumes of BC [65]. Its high-yield capacity makes it a staple in industrial BC production. Similarly, *K. hansenii* produces highly crystalline cellulose, while *K. sucrofermentans* efficiently metabolizes sucrose and fructose [16,66]. Although *Rhizobium* and *Agrobacterium* synthesize cellulose in natural contexts like root nodules or plant attachment structures [67,68], current evidence indicates much lower yields relative to *Komagataeibacter*. Among Gram-positives, *S. ventriculi* can generate cellulose-like biopolymers under extreme anaerobic and acidic conditions, yet its low productivity and demanding growth requirements limit its industrial potential [42,69]. Industrial optimization involves low-shear bioreactors and co-culture approaches. In kombucha, yeast-produced ethanol fuels BC synthesis by *Acetobacteraceae* [70,71]. Co-culturing *K. xylinus* with *Bacillus cereus* can boost yields ~3.7-fold via metabolites such as acetoin and 2,3-butanediol, which enhance respiration and ATP generation [72]. It is important to note that BC is also a by-product of kombucha fermentation, where the symbiotic culture of bacteria and yeast (SCOBY) plays a central role [73]. In this system, yeasts hydrolyze sucrose into glucose and fructose and produce ethanol, which is oxidized by acetic acid bacteria into acetic acid, thereby creating favorable conditions for BC synthesis. Recent studies have sought to restore and control this symbiotic relationship in laboratory and industrial contexts to enhance BC yield and quality [74,75]. For instance, optimizing the ratio of *Komagataeibacter intermedius* with yeasts such as *Brettanomyces bruxellensis* and *Zygosaccharomyces bisporus* achieved yields up to 5.51 g/L dry basis [72]. Similarly, exploration of *Acetobacteraceae* strains in kombucha SCOBY samples revealed that certain strains perform better under stirred fermentation, producing significantly higher amounts of BC [71]. These findings highlight the potential of microbial consortia not only in kombucha-based systems but also in scalable industrial strategies for efficient BC production from agro-industrial substrates. In summary, while diverse microorganisms are capable of cellulose biosynthesis, recent research continues to reinforce the industrial superiority of *Komagataeibacter* species for scalable and efficient BC production.

#### Synthetic Biology for Functional Modification of BC

Synthetic biology has emerged as a transformative avenue for engineering BC with enhanced and novel functionalities. In particular, *Komagataeibacter* spp. have become model organisms in this realm, owing to their robust cellulose-producing capability and amenability to genetic modification [76]. This approach transcends traditional media optimization by integrating engineered biosynthetic modules into BC-producing strains, enabling the in situ incorporation of functional molecules during cellulose synthesis [76,77]. Recent advancements include the modular reprogramming of *Komagataeibacter* cells using synthetic biology principles such as standardized promoters, vectors, and regulatory elements to fine-tune BC production and introduce tailored physicochemical traits [77]. Additionally, the concept of engineered living materials has gained traction: BC is being developed as a programmable, responsive biomaterial, capable of dynamic adaptation and high-value performance across applications ranging from textiles to bioelectronics [78]. Furthermore, synthetic biology-driven microbial consortia are being explored to construct BC-based composite materials, where *Komagataeibacter* partners with heterologous microbes to enable simultaneous cellulose synthesis and functionalization in a co-cultivation system [79]. Such composite living systems hold potential for the development of multifunctional biomaterials with integrated material and enzymatic functions. Finally, high-throughput strategies such as directed evolution of *Komagataeibacter sucrofermentans* under synthetic biology frameworks have led to the discovery of genetic variants with enhanced BC overproduction capabilities, unveiling unexpected genetic targets for future synthetic optimization [80].

## 3. Structure and Properties of BC

### 3.1. Structural Characteristics of BC

BC is a biomaterial characterized by a three-dimensional, network-like structure composed of nanometer-scale cellulose fibers. Cellulose-producing bacteria, such as *Komagataeibacter*, secrete polysaccharide chains into the extracellular environment, where they self-assemble into hydrogel-like pellicles [81]. Each fiber has a diameter of only a few tens of nanometers (typically <100 nm), making it approximately one hundred times thinner than plant-derived cellulose fibers [82]. These ultra-thin nanofibers, which can reach several micrometers in length, form a highly interconnected mesh with a microporous architecture. The average pore size between the fibers ranges from ~10 to 300 nm [83]. In contrast, plant-derived cellulose is embedded in a complex matrix with lignin and hemicellulose, forming microfibers of 0.2–0.5 µm in diameter [84,85]. BC nanofibers typically range from 5 to 100 nm, with most in the 5–10 nm range under optimized conditions using low-cost substrates from agricultural waste (Figure 2) [86]. At the molecular level, cellulose is composed of glucose units linked by *β*-1,4-glycosidic bonds [87]. Each glucose has three hydroxyl (–OH) groups that allow hydrogen bonding between chains. In BC, these chains are tightly packed due to inter- and intramolecular hydrogen bonds, forming strong microfibrils [88,89]. Van der Waals interactions add to the axial stiffness and mechanical stability of the polymer [7]. Natural cellulose mostly exists as cellulose I, which includes two forms: *Iα* and *Iβ* [90,91]. BC predominantly exists as the *Iα* form, while plant cellulose is mostly *Iβ* [92,93]. *Iα* is less thermodynamically stable and can convert to *Iβ* over time or upon heat treatment [94].

BC can also be converted to cellulose II using alkaline treatments such as sodium hydroxide [95,96]. Despite this, BC exhibits a high degree of crystallinity (84–89%) compared to 40–60% in plant cellulose [97]. This high crystallinity promotes intermolecular interactions, resulting in enhanced strength, chemical resistance, and structural stability. BC fibers grow in random directions, forming a robust 3D network. At fiber junctions, hydrogen bonds maintain the integrity of the matrix. Thanks to this architecture, the BC hydrogel retains its shape and mechanical properties even when fully hydrated [98,99]. These unique properties make BC an ideal candidate for sustainable material development through microbial valorization of agro waste, although in most cases it requires further modification or functionalization to tailor its properties for specific applications, contributing to circular bioeconomy goals.

### 3.2. Functional and Physical Properties of BC

The key physical properties of BC include high mechanical strength, flexibility, and excellent water-holding capacity [100]. Since BC consists of pure cellulose, it exhibits remarkable tensile strength. For example, dry BC films can withstand forces ranging from 200 to 300 MPa. In comparison, plant-derived cellulose fibers such as cotton or flax may reach tensile strengths of 750–1000 MPa; however, these fibers often contain additional components and have a more rigid, less flexible structure [101,102]. It should be noted that tensile strength values of BC are not constant and may vary significantly depending on the fermentation method used. Generally, static culture tends to yield denser pellicles with higher tensile strength, whereas agitated culture provides higher volumetric yields but often results in BC with lower or less uniform mechanical properties.

BC nanofibers are randomly oriented, forming an isotropic, three-dimensional network, which provides uniform mechanical properties in all directions [103]. This nanostructure imparts softness and shape adaptability, especially in hydrated conditions. When hydrated, BC retains its shape well and can form a thin film that conforms to any surface making it highly effective as a biocompatible wound dressing [104].

BC also demonstrates exceptional water retention, capable of absorbing up to 100–200 times its dry weight in water [105]. Studies show that freshly synthesized BC hydrogel contains ~98–99% water, with approximately 90% of that being bound water within the nanofiber network. This bound water is essential for maintaining gel structure, elasticity, and biofunctionality [106]. In addition, BC displays high vapor permeability (~2900 g/m^2^/day), allowing it to sustain a moist environment over skin or food surfaces without causing dehydration [107,108].

### 3.3. Chemical Properties and Modification Potential of BC

Chemically, BC is considered one of the purest forms of natural cellulose [109]. Unlike plant-based sources, BC is synthesized directly as a high-purity polymer, without the need for extensive purification steps [110]. Due to its biosynthetic origin, BC forms a highly crystalline, easily modifiable matrix, ideal for functionalization. Each glucose unit contains three hydroxyl (–OH) groups that are exposed on the surface [111]. These groups can be targeted to attach various chemical compounds, allowing new functional properties to be introduced. For example, antiseptics, dyes, or other polymers can be covalently bound to the surface of BC, resulting in new types of functional materials [112]. Its lack of interfering biopolymers like lignin makes BC an ideal substrate for controlled chemical modifications. In contrast, plant cellulose must first undergo complex pre-treatment steps to remove lignin, waxes, and other impurities before it can be modified [113,114]. Chemical modification of BC can be performed either post-synthetically (via esterification, oxidation, or grafting) or in situ, during microbial fermentation, by incorporating functional molecules into the culture medium. These methods expand the material’s utility in fields such as drug delivery, biosensing, and smart packaging [115]. Its inherent hydrophilicity and surface charge allow for controlled interactions with bioactive agents, making it a favorable platform for biomedical applications. In addition to its modification potential, BC possesses intrinsic chemical properties that enhance its functionality. Its high crystallinity with cellulose *Iα* allomorph provides strong hydrogen bonding networks, ensuring chemical resistance and thermal stability [116]. BC is stable under mild acidic and alkaline conditions but can be hydrolyzed by strong acids or degraded by cellulases. The hydroxyl-rich surface confers hydrophilicity, high sorption capacity, and reactivity toward esterification, etherification, and oxidation. These features influence both its baseline performance and the efficiency of subsequent functionalization processes [117,118].

### 3.4. Biomedical Properties of BC

BC is a biocompatible, biologically inert, and non-toxic material, making it highly suitable for contact with human tissues and biomedical applications [119,120]. Studies have confirmed that human skin cells such as fibroblasts and keratinocytes adhere, grow, and proliferate well on BC membranes without causing adverse reactions [121]. Even during long-term implantation, BC elicits minimal immune response, with low levels of inflammation and foreign body reactions [122]. Due to its excellent moisture retention, gas permeability, and structural flexibility, BC has been successfully applied in various biomedical fields, such as wound dressings, artificial skin for burn treatment, as well as cartilage scaffolds and vascular grafts [123]. The porous nanofibrillar network of BC supports tissue regeneration by maintaining a moist wound environment, preventing bacterial invasion, and facilitating oxygen exchange. In addition to its biomedical safety, BC is biodegradable under natural conditions. In the environment, various fungi and bacteria produce cellulase enzymes that can hydrolyze BC into glucose units, enabling complete biodegradation without toxic residues [124,125]. Unlike plant cellulose, which degrades slowly due to its lignin content, BC’s high purity accelerates natural decomposition [126]. Moreover, BC obtained through microbial valorization of agro-waste has shown consistent biocompatibility profiles, supporting its integration into bio-based medical products that align with circular bioeconomy principles [127].

## 4. Valorization of Agro-Industrial Waste into Bacterial Cellulose 

Agricultural and agro-industrial residues such as fruit peels, sugarcane bagasse, cereal straws, and corn stover have emerged as low-cost feedstocks for microbial production of BC in recent years. Utilizing these wastes not only addresses disposal issues but also reduces the cost of BC fermentation media [128]. Fruit processing wastes are high in sugars and have been widely explored as carbon sources for BC. Citrus peels and pomace can produce significantly more BC than standard sugar media, with Fan et al. reporting ~5.7 g/L BC from enzymatically hydrolyzed citrus waste, about 1.5× the yield on Hestrin–Schramm (HS) medium [129]. More recently, mango peel hydrolysate was used as a sole medium by an *Achromobacter* isolate, achieving 1.22 g/L BC after optimization [130]. Mixed fruit residues have also been tested: for example, a *Komagataeibacter medellinensis* strain grew on apple peel combined with sugarcane juice, yielding up to ~2.5 g/L BC in 14 days without expensive supplements [128]. *Waste figs* (a sugar-rich fruit byproduct) were shown to support one of the highest BC titers reported—8.45 g/L—when medium components (initial pH ~6.0, sugar ~63 g/L) were optimized via RSM [131]. These examples illustrate that fruit and vegetable wastes (orange, mango, fig, etc.) can be effectively valorized into BC, often yielding several grams per liter under optimized conditions. Sugarcane bagasse, a fibrous lignocellulosic residue, has been successfully converted to BC after hydrolysis. Akintunde et al. used enzymatically hydrolyzed bagasse to feed two novel *Komagataeibacter* strains, obtaining BC yields up to 1.2 g/L (dry weight) under continuous agitation [132]. In that study, agitation markedly improved yields compared to static culture (by ~4×, from 0.3 g/L on HS medium to 1.2 g/L on bagasse media). Lignocellulosic straw residues (rice, wheat, corn stover) require pretreatment to release fermentable sugars for BC production. *Rice straw* has been particularly studied. Sharma et al. (2025) pretreated rice straw with 2% NaOH and enzymatic saccharification, yielding a hydrolysate with ~29 g/L sugars [133]. Although specific data on wheat straw BC production in the recent literature are sparse, wheat straw (which contains ~35–45% cellulose) is also being evaluated after delignification [134]. Overall, these studies affirm that crop residues can be transformed into BC, provided that effective pretreatment and fermentation strategies are in place (Table 1).

Taken together, the examples summarized above reveal clear relationships between substrate type, pretreatment severity, and BC productivity. Fruit- and vegetable-derived residues (orange peel, mango peel, waste figs) generally yield higher BC titers (≈1.2–8.45 g/L) with minimal pretreatment, whereas lignocellulosic materials (rice straw, wheat straw, sugarcane bagasse) require more intensive alkaline or enzymatic pretreatment and result in moderate yields (≈1.2–7.17 g/L). Agro-industrial effluents such as molasses, corn steep liquor, and kombucha broth offer consistently high productivity due to readily available fermentable sugars. Most high-yield processes rely on *Komagataeibacter* spp., though certain non-acetic acid bacteria (e.g., *Achromobacter* sp. S3) perform competitively on specific waste streams like mango peel. Collectively, these comparative trends indicate that sugar-rich food-processing wastes are better suited for decentralized low-cost BC production, whereas lignocellulosic residues become more advantageous when integrated into biorefineries capable of offsetting pretreatment demands.

## 5. Process Optimization Strategies in BC Production

The choice and enhancement of the microbial producer is another pillar of process optimization. BC production is primarily carried out by *Komagataeibacter* species, the most efficient bacterial cellulose producers. However, wild-type strains can suffer from suboptimal yields, especially on unconventional substrates, due to incomplete sugar utilization or sensitivity to inhibitors. Recent efforts have focused on strain adaptation and genetic engineering to improve BC productivity [93,136]. For example, Anguluri et al. subjected a *K. xylinus* strain to adaptive laboratory evolution (ALE) on mannitol—a sugar alcohol present in some plant wastes—for 210 days. The evolved strain showed a 38% higher BC yield on mannitol (and similarly improved on fructose) compared to the starting culture [137]. This demonstrated that gradually acclimatizing bacteria to alternative carbon sources can enhance their metabolism and cellulose output. In another study, Rezaei et al. applied UV mutagenesis and ALE to a vinegar-isolated *Komagataeibacter*, achieving a sixfold increase in BC production (up to 9.3 g/L) versus the parent strain’s 1.3 g/L. Such mutant or adapted strains often exhibit higher tolerance to byproducts and overproduce cellulose [138]. Rational genetic engineering is also being applied. Montgomery-Silva et al. engineered a *K. sucrofermentans* by knocking out genes for pathways that divert glucose to byproducts like gluconic acid. The engineered strain produced 5.77× more BC than wild type on glucose, by preventing acid byproduct formation [139].

### 5.1. Genetic Engineering Approaches to Improve Cellulose Yield

Genetic engineering has emerged as a powerful strategy to overcome the intrinsic limitations of native *Komagataeibacter* strains, which, despite their high cellulose-producing capacity, often exhibit suboptimal yields when cultivated on complex or low-cost substrates. Recent advances in genomics and systems biology have pinpointed key bottlenecks in BC biosynthesis, such as limited precursor supply, carbon flux imbalances, and inefficient *bcs* operon regulation [140,141]. By applying targeted modifications ranging from single-gene overexpression to multi-gene metabolic rewiring, researchers have been able to redirect carbon flow, enhance enzyme activity, and expand the spectrum of utilizable carbon sources [142]. Recent engineering of *Komagataeibacter* spp. has enhanced the conversion of agricultural waste–derived sugars into cellulose [143,144]. One key approach is the overexpression of crucial BC biosynthetic genes for example, the *bcs* operon encoding the cellulose synthase complex along with enzymes that boost precursor supply, which significantly increases cellulose production [145,146]. In parallel, CRISPR-Cas9 genome editing and CRISPR interference (CRISPRi) have been employed to delete or repress genes that divert carbon into unwanted by-products for instance, knocking out glucose dehydrogenase to block gluconic acid formation thereby channeling more substrate toward cellulose synthesis and yielding higher BC titers [147,148]. Metabolic engineering and synthetic biology approaches have also expanded substrate utilization and improved regulatory control: introducing heterologous pathways enables the assimilation of pentose sugars from lignocellulosic hydrolysates, while overexpression of diguanylate cyclases elevates intracellular c-di-GMP levels to activate cellulose synthase, both of which further enhance BC productivity [149,150]. Collectively, these genetic modifications have achieved substantial gains in BC output often reporting several-fold increases in titer or yield in engineered *Komagataeibacter* strains cultivated on agricultural waste feedstocks, underscoring innovative routes for integrating BC production into a circular bioeconomy [146,151].

### 5.2. Fermentation Condition Optimization

Once suitable feedstock and strains are in place, fine-tuning the fermentation conditions is crucial. Recent research heavily employs statistical design of experiments (DoE) to optimize factors like pH, temperature, carbon/nitrogen ratio, and additives for BC production from wastes. pH control is particularly important: *Komagataeibacter* tends to oxidize sugars to organic acids (e.g., gluconic acid) which drop the pH and can slow BC synthesis [152]. Maintaining the pH in a favorable range (often 5.5–7.0) prevents this feedback inhibition. For instance, Yilmaz et al. found an initial pH ~6.0 optimal for waste fig medium [131], while Sharma et al. similarly noted pH 6 gave the best results for rice straw hydrolysate. Temperature is typically kept around 28–30 °C for mesophilic BC producers; slight optimization can improve yields [133].

Furthermore, aeration and agitation are key parameters in fermentation. BC synthesis is strictly aerobic, so oxygen transfer can be limiting in static cultures [153]. Agitating the culture or sparging air can enhance oxygen availability but must be balanced against shear that disrupts the cellulose network. Optimization of agitation speed has been reported: moderate shaking or intermittent agitation often gives higher BC output than fully static conditions [153].

### 5.3. Omics Technologies and Advanced Strain Development

Omics technologies including genomics, transcriptomics, and proteomics have become pivotal tools for improving BC producing microorganisms. Recent genome-based investigations revealed that the newly characterized *Komagataeibacter uvaceti* FXV3 produces up to three times more BC than *K. xylinus* and harbors unique genes linked to enhanced stress tolerance [154]. Recent comparative genomic analyses have examined the genomes of 79 different BC-producing strains. These studies revealed various structural variations in the *bcs* operon responsible for cellulose biosynthesis as well as differences in carbohydrate metabolism pathways. Such findings help explain the diversity in productivity and functional properties observed among different BC-producing bacteria [155]. The availability of complete genome sequences has facilitated targeted strain improvement strategies. CRISPR/Cas9-based genome editing, along with advanced base-editing systems, enables simultaneous modification of multiple genes to redirect carbon flux toward cellulose biosynthesis. For instance, Xin et al. developed a CRISPR-guided cytosine base editor capable of efficiently modifying three genes at once, demonstrated by inactivating genes involved in mannitol metabolism to improve BC yield [156]. Integration of omics datasets with substrate utilization profiling from agricultural waste hydrolysates now allows researchers to identify metabolic bottlenecks and regulatory targets for optimization. This combined approach accelerates the development of robust, high-yield, and stress-tolerant *Komagataeibacter* strains, thereby supporting the sustainable production of eco-friendly cellulose-based biomaterials for applications in food packaging, biomedicine, and other sectors of the circular bioeconomy.

### 5.4. Bioreactor Engineering and Scale-Up Strategies

Translating laboratory successes to industrial-scale BC production requires suitable bioreactor designs that maintain high productivity and quality [157]. Traditional BC fermentation in static trays yields a thick pellicle with excellent properties but has low volumetric productivity. Agitated tank reactors, on the other hand, increase yield rate but often produce fragmented BC or cellulose in dispersed forms. To bridge this gap, researchers are developing specialized bioreactors for BC. One promising approach is the Rotating Disk Bioreactor (RDB), which periodically exposes growing cellulose biofilms to air by rotating them in and out of the liquid medium [158]. Cáceres et al. constructed an RDB and showed that using rough, silicone-coated disks at 7–9 rpm significantly increased BC production (up to 2.72 g/L in 12 days) compared to static culture using a standard synthetic medium [158]. By contrast, Xu et al. reported one of the most successful scale-up attempts with a 15 L bioreactor operated on rice bran-derived medium (an agro-industrial waste substrate), achieving 20.7 g/L BC after ~2 weeks. This clearly demonstrates the feasibility of agro-waste valorization in controlled bioreactor systems [45]. Future research should further explore specialized bioreactor designs tailored for agro-industrial waste substrates, as their heterogeneity and variable composition may affect oxygen transfer, mixing, and nutrient availability. Developing modular, low-shear reactors and continuous feeding strategies could improve scalability. Additionally, integrating pretreatment and fermentation steps into closed loop biorefinery concepts will be critical for maximizing yield and ensuring industrial feasibility. In conclusion, advances in pretreatment, microbial engineering, process optimization, and reactor design have improved the conversion of agricultural wastes into high-value BC within a circular bioeconomy. Ongoing research aims to further boost yields and expand usable waste types, bringing sustainable large-scale BC production closer to reality.

## 6. Circular Bioeconomy Applications of BC Production

In recent years, the concept of a circular bioeconomy aimed at recycling waste and conserving natural resources has gained considerable traction. Within this framework, BC plays a significant role as a biopolymer with a clean structure, produced through environmentally friendly microbial biosynthesis that requires no harsh chemical treatments [159]. BC is a highly crystalline, strong, and biocompatible material. However, its production cost remains a major bottleneck, particularly due to the carbon source used. Studies have shown that the growth medium can account for 30–40% of the total BC production cost [160,161]. While synthetic media (e.g., glucose-based HS medium) yield high BC output, they are economically unsustainable in large-scale production, costing up to 40 EUR/kg [162]. In contrast, numerous studies have reported that using agro-industrial wastes such as fruit peels, sugarcane bagasse, molasses, and wheat straw hydrolysates as low-cost substrates can reduce medium costs by 50–70% without compromising BC yield or quality [163,164]. This significant cost reduction directly supports the circular bioeconomy paradigm by turning underutilized residues into high value bioproducts. In addition to substrate-related strategies, the use of mixed microbial cultures (MMCs) has also been proposed as a potential approach to further reduce BC production costs. MMCs can improve process robustness, operate under non-sterile or semi-sterile conditions, and utilize low-grade or variable waste substrates, thereby lowering sterilization and operational expenses in large-scale systems. For comparison, conventional polymers such as polyethylene terephthalate (PET) and polylactic acid (PLA) are produced at substantially lower costs, typically ranging from 1.2 to 2.0 EUR/kg for PET and 2.5 to 3.0 EUR/kg for PLA, while plant-derived cellulose can be extracted at approximately 0.5–1.0 EUR/kg. However, a more appropriate comparison for BC is polyhydroxyalkanoates (PHA), as both BC and PHA are microbially synthesized, biobased, and biodegradable polymers sharing the “three-times bio” characteristics [165,166]. Techno-economic analyses place industrial-scale BC costs in the ≈14.8–63.8 USD/kg range, with some optimized scenarios near ≈30 USD/kg; adopting agro-industrial residues to cut medium expenses is a realistic route to further reduce costs [167,168,169,170]. Common feedstocks such as fruit and vegetable residues, sugar industry by-products, and winemaking waste are rich in fermentable sugars and can be used to cultivate *Komagataeibacter xylinus*, producing BC in both static and agitated culture systems [171]. These processes form hydrated networks that are further processed into functional films, fibers, coatings, and bio-composites [172]. Aligned BC nanofibers produced through methods like uniaxial stretching, shear alignment, magnetic-assisted assembly, and directional freezing have recently attracted attention due to their enhanced tensile strength and cell guidance properties [173,174]. These functionalizations expand BC’s use in advanced bio-based materials. Waste-derived BC materials contribute to diverse high-value applications, complementing the broader technological uses outlined earlier [175,176]. Notably, BC-based textiles offer a biodegradable alternative to synthetic fabrics, reducing plastic pollution [177]. After use, BC materials naturally decompose in the environment. The primary polymer, cellulose, is broken down by microbial enzymes into glucose and reintegrated into natural nutrient cycles [178]. Thus, the entire process from raw material to product disposal forms a closed loop, aligning with the principles of a sustainable circular bioeconomy (Figure 3).

In countries like Spain and Greece, wine-processing residues are used for BC production, reducing waste and energy inputs [179,180]. In Japan, BC is utilized in “nata de coco” desserts, exemplifying food-grade BC use in near-zero-waste industries [181]. Brazilian researchers have demonstrated BC-based medical films using fruit and dairy waste with high biocompatibility [182] further expanding the scope of circular economy applications. These examples demonstrate that BC products naturally degrade without harming the environment, effectively completing the circular loop. BC production supports a full material cycle from raw biomass to high-value biomedical and industrial materials, all the way to eco-friendly degradation [183]. One of the key advantages of BC implementation is its potential to replace plastics and synthetic materials, contributing to cost savings and reduced ecological impact [184]. Recent studies emphasize that industrial integration of BC is a promising pathway for realizing the principles of a circular bioeconomy [185,186]. Life Cycle Assessment (LCA) studies underscore the environmental advantages of BC derived from waste. One study reported cradle-to-gate CO_2_ emissions as low as ~39 kg CO_2_-eq/kg for waste-derived BC, significantly lower than for synthetic BC (~296 kg CO_2_-eq/kg) [187]. Additionally, BC production has shown lower environmental impacts than plant-based nanocellulose, especially when agro-waste substrates are used [188]. While these results demonstrate BC’s sustainability potential, comprehensive LCA studies directly comparing BC to plant cellulose across full life-cycle stages remain limited and are needed to validate these findings. In addition, future LCAs should also compare BC with other microbially derived biopolymers such as PHA, which represents a scientifically relevant reference point due to similarities in biosynthesis, biodegradability, and biobased origin.

Techno-economic evaluations further highlight that the overall feasibility of BC production depends on harmonizing process productivity with substrate cost, pretreatment intensity, and energy inputs. Integrating BC fermentation into broader circular bioeconomy and biorefinery systems such as linking waste-derived sugar streams with bioethanol, enzyme, or organic acid production can substantially reduce operating costs while improving resource efficiency. When combined with sustainability metrics, recent LCA data indicate that waste-derived BC can achieve markedly lower environmental footprints compared with both synthetic polymers and plant-derived nanocellulose, especially when chemical use, water consumption, and energy inputs are minimized. These insights collectively suggest that aligning BC production with integrated waste-to-value systems is essential for improving techno-economic performance and realizing its full potential within a circular bioeconomy.

## 7. Safety, Regulatory Compliance, and Quality Assurance in Waste-Derived BC Production

The utilization of agro-industrial and agricultural waste streams as substrates for BC production requires careful attention to safety, regulatory compliance, and consistent quality assurance, particularly when the final material is intended for food, biomedical, cosmetic, or packaging applications. Waste-derived feedstocks may contain a wide spectrum of contaminants, including microbial pathogens, pesticide residues, mycotoxins, heavy metals, polycyclic aromatic hydrocarbons (PAHs), or fermentation inhibitors, all of which can compromise the safety profile of the final BC product if not adequately controlled [160,164]. Therefore, rigorous pretreatment, sterilization, and compositional monitoring of waste streams are essential to ensure substrate suitability and to prevent the introduction of hazardous impurities into the biosynthesis process [164].

From a regulatory standpoint, food-grade BC must comply with requirements set by major international agencies, such as the U.S. Food and Drug Administration (FDA), which recognizes BC as Generally Recognized as Safe (GRAS) for direct food contact applications [181]. For biomedical uses including wound dressings, scaffolds, and implantable materials strict adherence to ISO 10993 standards [182] on biocompatibility, cytotoxicity, endotoxin limits, and sterility validation is mandatory to ensure patient safety [183]. Furthermore, European regulatory frameworks, such as EFSA and REACH, emphasize traceability, contaminant control, and the absence of harmful residues in bio-based materials intended for consumer markets [166].

Quality assurance challenges arise due to the natural variability of waste substrates, as their chemical and nutritional composition can fluctuate depending on season, crop variety, processing conditions, and storage time. These fluctuations can lead to inconsistent BC yields, altered nanostructure, or undesired by-products. Therefore, implementing standardized quality control protocols such as batch-to-batch characterization, contaminant screening, heavy metal analysis, and monitoring of pH, carbon-to-nitrogen ratio, and fermentation kinetics—is essential to ensure reproducible BC quality [170,171]. Additionally, safety validation measures, including microbial load assessment, endotoxin testing, and residual substrate analysis, are necessary to guarantee the integrity and purity of the final product [172].

LCA studies further highlight that the environmental and safety advantages of waste-derived BC are maximized only when pretreatment chemicals, water consumption, and energy use are minimized. For example, cradle-to-gate analyses demonstrate that BC synthesized from agro-waste exhibits significantly lower environmental impacts compared with synthetic polymers and plant-based nanocellulose, particularly when integrated into circular bioeconomy and biorefinery systems that valorize all residual streams [188,189,190]. These findings underscore that robust safety, regulatory, and quality assurance frameworks are critical not only for compliance but also for ensuring the long-term sustainability and market acceptance of waste-derived BC in high-value applications.

## 8. Conclusions

The microbial production of BC from agricultural and agro-industrial residues represents a sustainable route for converting low-value biomass into high-value materials, fully aligned with the principles of the circular bioeconomy. This review outlined recent advances in bioprocess strategies including optimized static and agitated fermentations, co-cultivation systems, diversified carbon sources, stepwise nutrient feeding, and the use of genetically engineered *Komagataeibacter* strains that collectively enhance BC yield, reduce production costs, and enable effective valorization of agro-industrial waste streams. Despite these advances, key challenges remain, including industrial-scale scalability, variability in feedstock composition, process economics, and maintaining consistent product quality. Addressing these will require the development of cost-effective, modular, and adaptable production platforms capable of processing diverse agricultural and agro-industrial residues with minimal pretreatment. Equally important is the design of BC-based functional composites and integration of BC production into closed-loop biorefinery models, ensuring full resource recovery and minimal environmental impact. If these technological and economic barriers are overcome, BC derived from agricultural and agro-industrial waste could play a transformative role in replacing petroleum-based materials, contributing both to environmental protection and to the expansion of a resilient, circular bioeconomy. Nevertheless, compared with conventional polymers such as PET, PLA, and plant-derived cellulose, BC remains costlier at the industrial scale. Valorization of agro-industrial residues thus represents a realistic strategy to narrow this gap and strengthen BC’s position as a sustainable alternative in the circular bioeconomy.

## Figures and Tables

**Figure 1 microorganisms-13-02686-f001:**
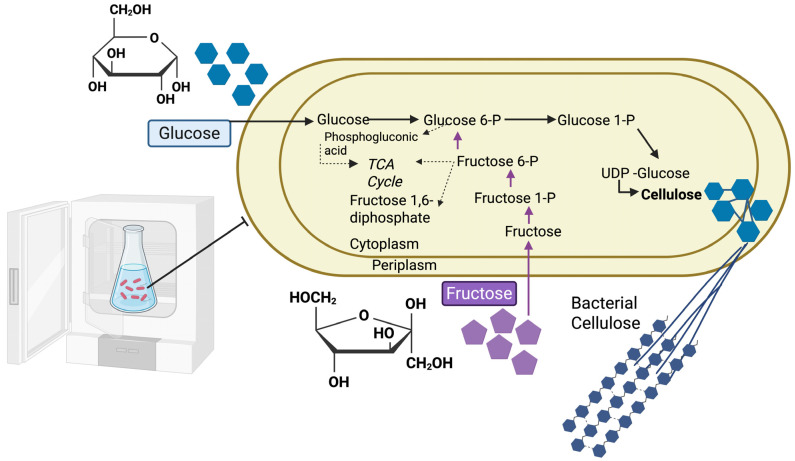
Metabolic pathway of BC biosynthesis. Glucose and fructose are taken up and phosphorylated into glucose-6-phosphate and fructose-6-phosphate, which feed into the pentose phosphate and glycolytic pathways. Glucose-1-phosphate is further activated to UDP-glucose, the direct substrate for cellulose synthase. UDP-glucose is polymerized into β-1,4-glucan chains and exported across the cell envelope, where they assemble into crystalline BC microfibrils. This pathway highlights how carbon metabolism is redirected toward polymer synthesis, linking primary sugar utilization to sustainable biopolymer production. Note: Created with BioRender.com, License No. XC28LV5FD3 (accessed on 1 November 2025).

**Figure 2 microorganisms-13-02686-f002:**
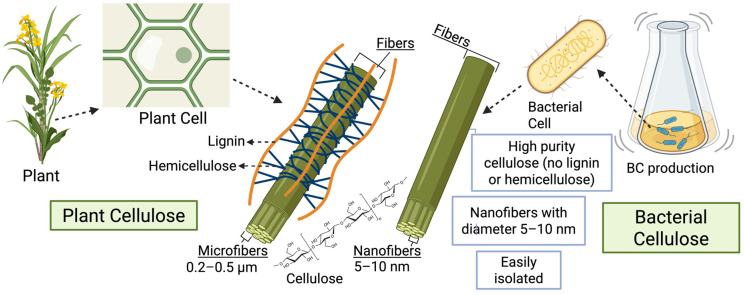
Structural and morphological comparison between plant-derived and BC. Plant cellulose is associated with lignin and hemicellulose, forming larger microfibers (0.2–0.5 µm) that require pretreatment for isolation. In contrast, BC is synthesized as highly pure nanofibers (5–10 nm) without lignin or hemicellulose, directly suitable for structural and functional applications. Note: Created with BioRender.com, License No. EP28LEOS6U (accessed on 1 November 2025).

**Figure 3 microorganisms-13-02686-f003:**
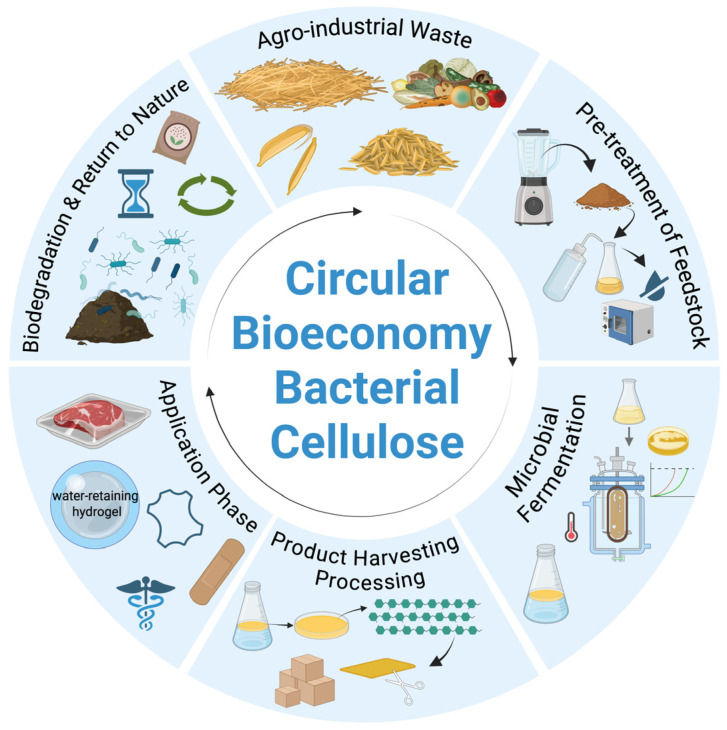
Circular cycle of BC production from agricultural and agro-industrial waste. Pretreated residues provide fermentable sugars that are converted into BC via microbial fermentation. The harvested BC is processed into films, hydrogels, and composites for packaging and biomedical applications, which eventually biodegrade and return to nature, completing the circular bioeconomy loop. Note: Created with BioRender.com, License No. AF28FRFLY1 (accessed on 1 November 2025).

**Table 1 microorganisms-13-02686-t001:** BC production from agro-industrial residues—substrates, strains, pretreatments, and yields.

Waste Substrate	Pretreatment	Microbial Strain	BC Yield	Fermentation Time	Productivity (g/L·h)	Pretreatment Complexity	Application	Ref.
Mango peel waste	Nitric acid hydrolysis	*Achromobacter* sp. S3	1.22 g/L (optimized)	~7 d (168 h)	0.0073	Medium	Low-cost BC films	[130]
Orange peel & pomace	Enzymatic hydrolysis (yeast extract	*K. xylinus* CICC 10529	5.7 ± 0.7 g/L	~12 d (288 h)	0.0198	Medium	Food packaging, hydrogels	[129]
Waste figs (discard)	Blended (natural sugars)	*K. xylinus* (RSM optimized)	8.45 g/L	~14 d (336 h)	0.0251	Low	High-strength BC membranes	[131]
Sugarcane bagasse	Enzymatic hydrolysate	*Komagataeibacter* sp.	1.2 g/L (agitated)	~10 d (240 h)	0.0050	Medium	Biorefinery integration	[132]
Corncob (corn stover)	Enzymatic hydrolysate	*Komagataeibacter* sp.	1.6 g/L (agitated)	~10 d (240 h)	0.0067	Medium	Biofilms, composites	[132]
Rice straw	2% NaOH + enzymatic hydrolysis	*K. xylinus* (optimized)	7.17 ± 0.05 g/L	~12 d (288 h)	0.0249	High	Biomedical-grade BC	[133]
Rice bran (scale-up)	Autoclave (nutrient-rich)	*K. europaeus* (15 L)	20.7 g/L (15 d)	15 d (360 h)	0.0575	Low	Industrial BC production	[79]
Asparagus peel	Acid/enzyme hydrolysate	*K. rhaeticus* QK23	2.57 g/L (25 d)	25 d (600 h)	0.0043	High	Composite applications	[135]

## Data Availability

No new data were created or analyzed in this study. Data sharing is not applicable to this article.

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
