# Peer review of "Microbial Valorization of Agricultural and Agro-Industrial Waste into Bacterial Cellulose: Innovations for Circular Bioeconomy Integration"

_microorganisms, 2025, doi:10.3390/microorganisms13122686_

Round 1
Reviewer 1 Report
Comments and Suggestions for Authors
The manuscript “Microbial Valorization of Agricultural and Agro-Industrial Waste into Bacterial Cellulose: Innovations for Circular Bioeconomy Integration” presents a relevant and timely review on the use of microbial biotechnology for the valorization of agricultural residues. The paper is generally well-structured, scientifically sound, and well-aligned with the journal’s scope.
To further improve the manuscript, the following points are recommended:
- Provide a clearer description of the methodology applied to identify and analyze the reviewed literature (databases consulted, time range, search terms, and selection criteria).
- Expand the comparative evaluation of bacterial strains, substrates, and production efficiencies, instead of presenting examples in a mainly descriptive manner.
- Broaden the discussion regarding techno-economic feasibility and integration into circular bioeconomy frameworks, including references to sustainability metrics or Life Cycle Assessment (LCA) where applicable.
- Incorporate a brief section addressing issues of safety, regulatory compliance, and quality assurance related to the use of waste-derived substrates, especially for biomedical and food industry applications.
- Include a summary table or schematic that synthesizes the types of residues, producing strains, main process conditions, and applications of bacterial cellulose.
Overall, this is a valuable contribution to the field. With these minor revisions, the manuscript will present a more critical and comprehensive perspective.
Recommendation: Minor Revisions Required
Author Response
We sincerely thank the reviewer for the constructive and insightful comments. All suggestions have been carefully addressed, and corresponding revisions have been incorporated into the manuscript as detailed below.
Comments 1: Provide a clearer description of the methodology applied to identify and analyze the reviewed literature (databases consulted, time range, search terms, and selection criteria).
Response 1: Thank you for this valuable comment. A concise description of the literature search methodology has now been added to the revised manuscript. The new paragraph specifies the databases consulted (Web of Science, Scopus, PubMed, ScienceDirect, Google Scholar), the focus on recent publications, the main keyword combinations used, and the inclusion/exclusion criteria applied. In addition, earlier seminal studies were retained when they provided essential foundational insights relevant to bacterial cellulose biosynthesis and its mechanistic background (Lines: 114-124).
Comments 2: Expand the comparative evaluation of bacterial strains, substrates, and production efficiencies, instead of presenting examples in a mainly descriptive manner.
Response 2: Thank you for this comment. In Section 4, we added a comparative paragraph that synthesizes the differences in BC yields across various agro-industrial substrates and highlights strain-specific performance and pretreatment effects, thereby replacing the previously descriptive examples with a clearer analytical comparison (Lines: 691-703).
Comments 3: Broaden the discussion regarding techno-economic feasibility and integration into circular bioeconomy frameworks, including references to sustainability metrics or Life Cycle Assessment (LCA) where applicable.
Response 3: The discussion on techno-economic feasibility and integration into circular bioeconomy frameworks has been expanded in the revised manuscript. A new analytical paragraph was added to Section 6, addressing key cost determinants, the role of biorefinery integration, and sustainability considerations (Lines: 963-974).
Comments 4: Incorporate a brief section addressing issues of safety, regulatory compliance, and quality assurance related to the use of waste-derived substrates, especially for biomedical and food industry applications.
Response 4: Thank you for this important comment. In the revised manuscript, we have added a dedicated subsection that addresses safety considerations, regulatory compliance, and quality assurance issues related to the use of waste-derived substrates, particularly for food and biomedical applications. This new section discusses potential contaminants that may be present in agro-industrial residues, relevant regulatory frameworks, and the need for standardized quality control procedures to ensure the safety and consistency of waste-derived BC. Appropriate references have been incorporated to support this expanded discussion (Lines: 976-1015).
Comments 5: Include a summary table or schematic that synthesizes the types of residues, producing strains, main process conditions, and applications of bacterial cellulose.
Response 5: Thank you for this helpful suggestion. In the revised manuscript, Table 1 has been updated by adding an additional column summarizing the main process conditions and application relevance for each waste substrate (Lines: 674-694).

Reviewer 2 Report
Comments and Suggestions for Authors
Major comment 1 – Expansion of the introductory section
The introduction provides a clear overview of the microbial valorization of agro-industrial residues into bacterial cellulose (BC). However, I believe this section could be strengthened by broadening the discussion on the general state of the art concerning agricultural and agro-industrial waste management and the various strategies currently available for their valorization. At present, the authors focus exclusively on BC production, while other biotechnological conversion routes, such as the production of organic acids via acidogenic fermentation and their subsequent use for polyhydroxyalkanoates (PHA) synthesis, are not mentioned. Including a brief discussion on these complementary pathways would help readers gain a more comprehensive understanding of the current valorization landscape.
In this context, I suggest the authors consider citing the following relevant studies, which fit well within line 92, where biopolymer production is discussed:
- Marchetti et al., 2025 – Unlocking the potential of food industry by-products: Sustainable volatile fatty acids production via mixed culture acidogenic fermentation of reground pasta – Journal of Cleaner Production, 526, 146633. https://doi.org/10.1016/j.jclepro.2025.146633
- Marchetti et al., 2024 – Valorization of Reground Pasta By-Product through PHA Production with Phototrophic Purple Bacteria – Catalysts, 14, 239. https://doi.org/10.3390/catal14040239.
Major comment 2 – Reorganization of manuscript structure
The current structure of the manuscript is generally clear and well-organized; however, I would suggest reconsidering the order of Sections 2 and 3 to improve the logical flow of the text. At present, Section 2 discusses the structure and properties of bacterial cellulose (BC), while Section 3 focuses on its biosynthesis and microbial production. In my opinion, presenting the biosynthesis and production aspects first (currently Section 3) and then describing the structural and functional properties (currently Section 2) would result in a more coherent and logically ordered manuscript.
Additionally, I recommend moving Subsections 3.2.1 (“Genetic Engineering Approaches to Improve Cellulose Yield”) and 3.2.3 (“Omics Technologies and Advanced Strain Development”) to Section 5, where optimization strategies for BC production are discussed in more detail, including examples of genetic engineering approaches. Merging these subsections into Section 5, while avoiding redundancy, would strengthen the overall structure and reduce repetition within the manuscript.
Major comment 3 – Comparison of BC with appropriate biopolymers
In Section 6 “Circular Bioeconomy Applications of BC Production”, line 469, the manuscript presents a cost comparison between BC and conventional plastics such as PET and PLA. However, since BC is microbiologically synthesized, it would be more appropriate to compare it with another microbially derived biopolymer, such as polyhydroxyalkanoates (PHA), which shares similar characteristics, being biobased, biodegradable, and biologically produced (the so-called “three-times bio” concept).
This point could also be reinforced at line 510, where the authors write:
“…comprehensive LCA studies directly comparing BC to plant cellulose across full life-cycle stages remain limited and are needed to validate these findings.”
Here, including a reference to PHA as a potential comparison target would make the discussion more comprehensive and scientifically relevant.
Major comment 4 – Potential strategies to further reduce BC production costs
The authors provide a thorough discussion on the economic aspects and potential cost-reduction strategies for BC production. However, it could be interesting to mention whether the use of mixed microbial cultures (MMCs), instead of pure or genetically engineered strains,
has been explored or could represent a feasible alternative for lowering production costs. MMCs may offer advantages in terms of process robustness, substrate flexibility, and reduced sterilization requirements, which could align well with large-scale and low-cost production approaches.
Minor comment 1 – Title specificity of Section 4
Once the introductory section on agro-industrial waste valorization has been expanded, I recommend making the title of Section 4 more specific. For instance, instead of the current general title, something like “Valorization of Agro-Industrial Waste into Bacterial Cellulose (BC)” would better reflect the content and help readers immediately grasp the focus of the section.
Minor comment 2 – Redundancy and repetition throughout the text
Several concepts are reiterated multiple times throughout the manuscript. For example:
- Line 170: “This makes BC a highly pure polysaccharide composed entirely of glucose units” already explained in Section 2.1.
- Line 214: “BC is a linear extracellular polymer composed of glucose monomers linked by β-1,4-glycosidic bonds” is conceptually redundant with the previous statement.
- Line 398: “Most studies still utilize acetic acid bacteria of the genus Komagataeibacter, which are renowned BC producers” already discussed in Section 3.
- Line 484: “The resulting waste-derived BC products have been applied in agriculture, food packaging, biomedicine, and even in electronics and wearable devices” is also mentioned earlier.
- Line 402: typo [107,150]] (double closing brackets)
I kindly suggest that the authors carefully revise the text to reduce such redundancies and improve the conciseness and overall readability of the manuscript.
Minor comment 3 – Figure readability
The figures are overall clear, well-designed, and representative of the discussed concepts. However, I recommend slightly enlarging the text and labels in Figures 1 and 2 to improve readability, especially for readers viewing the manuscript in print or at reduced scale.
Author Response
We sincerely thank the reviewer for their thorough and constructive evaluation of our manuscript. The comments have significantly improved the clarity, structure, and scientific depth of the paper. Redundant statements were removed, typographical errors corrected, and figure readability improved. All suggested revisions have been carefully implemented in the updated manuscript, with detailed point-by-point responses provided below.
Major comment 1 – Expansion of the introductory section
The introduction provides a clear overview of the microbial valorization of agro-industrial residues into bacterial cellulose (BC). However, I believe this section could be strengthened by broadening the discussion on the general state of the art concerning agricultural and agro-industrial waste management and the various strategies currently available for their valorization. At present, the authors focus exclusively on BC production, while other biotechnological conversion routes, such as the production of organic acids via acidogenic fermentation and their subsequent use for polyhydroxyalkanoates (PHA) synthesis, are not mentioned. Including a brief discussion on these complementary pathways would help readers gain a more comprehensive understanding of the current valorization landscape.
In this context, I suggest the authors consider citing the following relevant studies, which fit well within line 92, where biopolymer production is discussed:
- Marchetti et al., 2025 – Unlocking the potential of food industry by-products: Sustainable volatile fatty acids production via mixed culture acidogenic fermentation of reground pasta – Journal of Cleaner Production, 526, 146633. https://doi.org/10.1016/j.jclepro.2025.146633
- Marchetti et al., 2024 – Valorization of Reground Pasta By-Product through PHA Production with Phototrophic Purple Bacteria – Catalysts, 14, 239. https://doi.org/10.3390/catal14040239.
Response 1: Thank you for this helpful comment. We have expanded the Introduction to briefly discuss additional valorization pathways beyond BC production, including acidogenic fermentation for VFA generation and their use in PHA synthesis. The suggested studies by Marchetti et al. (2025; 2024) have been incorporated, and a concise paragraph has been added near Line 92 to provide a broader overview of current waste-to-bioproduct strategies (Lines: 92-103).
Major comment 2 – Reorganization of manuscript structure
The current structure of the manuscript is generally clear and well-organized; however, I would suggest reconsidering the order of Sections 2 and 3 to improve the logical flow of the text. At present, Section 2 discusses the structure and properties of bacterial cellulose (BC), while Section 3 focuses on its biosynthesis and microbial production. In my opinion, presenting the biosynthesis and production aspects first (currently Section 3) and then describing the structural and functional properties (currently Section 2) would result in a more coherent and logically ordered manuscript.
Additionally, I recommend moving Subsections 3.2.1 (“Genetic Engineering Approaches to Improve Cellulose Yield”) and 3.2.3 (“Omics Technologies and Advanced Strain Development”) to Section 5, where optimization strategies for BC production are discussed in more detail, including examples of genetic engineering approaches. Merging these subsections into Section 5, while avoiding redundancy, would strengthen the overall structure and reduce repetition within the manuscript.
Response 2: We thank the reviewer for the helpful suggestion. We have reordered Sections 2 and 3, so that biosynthesis and microbial production now precede the description of BC structure and properties. Additionally, Subsections 3.2.1 and 3.2.3 have been relocated and merged into Section 5, where they fit better within the discussion of optimization strategies. Redundant content was removed to ensure a smooth and coherent structure. These changes have improved the overall logical flow of the manuscript (Lines: 125;408-636; 725-760; 779-807).
Major comment 3 – Comparison of BC with appropriate biopolymers
In Section 6 “Circular Bioeconomy Applications of BC Production”, line 469, the manuscript presents a cost comparison between BC and conventional plastics such as PET and PLA. However, since BC is microbiologically synthesized, it would be more appropriate to compare it with another microbially derived biopolymer, such as polyhydroxyalkanoates (PHA), which shares similar characteristics, being biobased, biodegradable, and biologically produced (the so-called “three-times bio” concept).
This point could also be reinforced at line 510, where the authors write:
“…comprehensive LCA studies directly comparing BC to plant cellulose across full life-cycle stages remain limited and are needed to validate these findings.”
Here, including a reference to PHA as a potential comparison target would make the discussion more comprehensive and scientifically relevant.
Response 3: We agree that PHA represents a more scientifically relevant comparison for BC, as both are microbially synthesized, biobased, and biodegradable biopolymers. Accordingly, we have revised Section 6 by adding a statement after line 469 (new line: 940) to highlight PHA as an appropriate benchmark for BC in terms of production cost and bio-based origin. In addition, we have expanded the discussion after line 510 (new line: 963-974) to emphasize that future LCA studies should also compare BC with PHA to provide a more comprehensive assessment of their relative sustainability. These additions improve the clarity and relevance of the manuscript.
Major comment 4 – Potential strategies to further reduce BC production costs
The authors provide a thorough discussion on the economic aspects and potential cost-reduction strategies for BC production. However, it could be interesting to mention whether the use of mixed microbial cultures (MMCs), instead of pure or genetically engineered strains,
has been explored or could represent a feasible alternative for lowering production costs. MMCs may offer advantages in terms of process robustness, substrate flexibility, and reduced sterilization requirements, which could align well with large-scale and low-cost production approaches.
Response 4: To address this point, we have added a brief statement in Section 6 noting that mixed microbial cultures (MMCs) may represent an additional strategy for reducing BC production costs. MMCs can enhance process robustness, utilize low-grade or variable substrates, and operate under non-sterile conditions, thereby lowering sterilization and operational expenses in large-scale production. This addition broadens our discussion of cost-reduction approaches and aligns with the reviewer’s recommendation (Lines: 853-858).
Minor comment 1 – Title specificity of Section 4
Once the introductory section on agro-industrial waste valorization has been expanded, I recommend making the title of Section 4 more specific. For instance, instead of the current general title, something like “Valorization of Agro-Industrial Waste into Bacterial Cellulose (BC)” would better reflect the content and help readers immediately grasp the focus of the section.
Response 1: We thank the reviewer for this helpful suggestion. In accordance with the recommendation, we have updated the title of Section 4 to “Valorization of Agro-Industrial Waste into Bacterial Cellulose” to more accurately reflect the section’s content and improve clarity for readers (Lines: 637).
Minor comment 2 – Redundancy and repetition throughout the text
Several concepts are reiterated multiple times throughout the manuscript. For example:
- Line 170: “This makes BC a highly pure polysaccharide composed entirely of glucose units” already explained in Section 2.1.
- Line 214: “BC is a linear extracellular polymer composed of glucose monomers linked by β-1,4-glycosidic bonds” is conceptually redundant with the previous statement.
- Line 398: “Most studies still utilize acetic acid bacteria of the genus Komagataeibacter, which are renowned BC producers” already discussed in Section 3.
- Line 484: “The resulting waste-derived BC products have been applied in agriculture, food packaging, biomedicine, and even in electronics and wearable devices” is also mentioned earlier.
- Line 402: typo [107,150]] (double closing brackets)
I kindly suggest that the authors carefully revise the text to reduce such redundancies and improve the conciseness and overall readability of the manuscript.
Response 2: Thank you for pointing out the redundant statements in the manuscript. We have revised all indicated instances to improve conciseness and avoid repetition. Specifically:
The sentence “BC is a highly pure polysaccharide composed entirely of glucose units” has been removed. The sentence “BC is a linear extracellular polymer composed of glucose monomers linked by β-1,4-glycosidic bonds” has been replaced with: “BC forms a characteristic extracellular nanofibrous network produced by Komagataeibacter species” (Lines:127-128).
The sentence “Most studies still utilize acetic acid bacteria of the genus Komagataeibacter, which are renowned BC producers” has been revised to:
“BC production is primarily carried out by Komagataeibacter species, the most efficient bacterial cellulose producers ” (Lines:707-708).
The sentence “The resulting waste-derived BC products have been applied in agriculture, food packaging, biomedicine, and even in electronics and wearable devices” has been replaced with: “Waste-derived BC materials contribute to diverse high-value applications, complementing the broader technological uses outlined earlier ” (Lines:875-876).
The typographical error “[107,150]]” has been corrected to “ new link [94,137]”.
Minor comment 3 – Figure readability
The figures are overall clear, well-designed, and representative of the discussed concepts. However, I recommend slightly enlarging the text and labels in Figures 1 and 2 to improve readability, especially for readers viewing the manuscript in print or at reduced scale.
Response 3: We thank the reviewer for this helpful suggestion. The text and labels in the figures have been enlarged to improve readability. Due to the reorganization of the manuscript sections, the original Figure 1 is now presented as Figure 2, and the original Figure 2 has been renumbered as Figure 1. The updated figures have been included in the revised manuscript (Lines:216; 563-564).

Round 2
Reviewer 2 Report
Comments and Suggestions for Authors
Thank you for making the requested changes and following the suggestions provided. The manuscript now appears clearer and more fluid to read, suitable for publication in the journal Microorganism.